# Sedentary behavior mediates the association between weight-adjusted waist index and cardiovascular disease in patients with diabetes

Zhihui Li[1], Min Wu[1], Peng Wang[1], Wei Xie[2], Yunqi Hua[3], Guo Shao[4‡]*, Hongwei Zhu[1‡]*

1 The Second Affiliated Hospital of Baotou Medical College, Inner Mongolia, Baotou, China, 2 Inner Mongolia Key Laboratory of Hypoxic Translational Medicine, Baotou Medical College, Inner Mongolia, Baotou, China, 3 Department of Medical Oncology, Baotou Cancer Hospital, Baotou, China, 4 Center for Translational Medicine, the Third People's Hospital of Longgang District, Shenzhen, China

☯ These authors contributed equally to this work.
‡ GS and HZ also contributed equally to this work.
* zhwyjs@163.com (HZ); shao.guo.china@gmail.com (GS)

## Abstract

### Background

Obesity and cardiovascular disease (CVD) are both associated with sedentary behavior. However, the role that sedentary behavior plays in the relationship between obesity and CVD in patients with diabetes remains unclear. This study aimed to examine how the weight-adjusted waist index (WWI) relates to CVD risk in patients with diabetes and to explore sedentary behavior's potential mediating role in this relationship.

### Methods

The present study analyzed data from 4,937 participants across seven NHANES cycles (2007–2020). Using logistic regression models and restricted cubic spline analyses, we evaluated the associations between WWI, sedentary behavior, and CVD in patients with diabetes. Subgroup and sensitivity analyses were conducted to confirm the stability of the results. Mediation analysis explored the mediating role of sedentary behavior in the associations between WWI and CVD.

### Results

The present study found that diabetic patients with CVD had higher WWI levels ($11.8 \pm 0.69 \frac{cm}{\sqrt{kg}}$)compared to diabetic patients without CVD ($11.60 \pm 0.73 \frac{cm}{\sqrt{kg}}$). WWI was positively associated with CVD in patients with diabetes(OR = 1.27, 95% CI: 1.06–1.53; P < 0.01), even after accounting for potential confounders. Sedentary behavior was also positively associated with CVD(OR = 1.04, 95% CI: 1.02–1.07; P < 0.01). In a restricted cubic spline analysis, we found that WWI and sedentary

**Data availability statement:** All relevant data are within the manuscript and its Supporting Information files.

**Funding:** This work was supported by: National Natural Science Fund (82060337) – G.S. Shenzhen City Science and Technology Plan Project Basic Research Surface Project (JCYJ20220531092412028 to G.S., JCYJ20230807121306012 to G.S.). Shenzhen Longgang District Medical and Health Science and Technology Plan Project (LGKCYLWS2021000033 to G.S., LGKCYLWS2023025 to G.S.). Natural Science Foundation of Inner Mongolia Autonomous Region (2025MS08169) – H.Z. Inner Mongolia Public Hospital Research Joint Fund Science and Technology Project (2024GLLH0605) – H.Z. Youth Science and Technology Talent Development Program Project of Baotou Medical College (BYJJ-QNGG 202417) – H.Z. Baotou City Health Science and Technology Project (wsjkkj2022057) – H.Z.

**Competing interests:** The authors have declared that no competing interests exist.

time were positively correlated and approximately linearly associated with CVD risk in patients with diabetes. As WWI levels and sedentary time increased, the risk of CVD increased in patients with diabetes. Mediating analysis revealed that sedentary behaviour mediated 13.43% of the association between WWI and CVD in patients with diabetes.

## Conclusion

We found that WWI is associated with an increased risk of CVD in patients with diabetes, and sedentary behavior partially mediates this relationship. Reducing sedentary behavior may be a key strategy to reduce obesity and CVD risk in patients with diabetes.

---

## Introduction

Diabetes is becoming more prevalent worldwide, with approximately 10.2% of the population estimated to suffer from it by 2030 [1]. As a major health challenge in the twenty-first century, diabetes has been recognized as a significant contributor to global disease burden [2]. Cardiovascular disease (CVD) is the leading cause of morbidity and mortality in diabetic patients [3]. Therefore, the primary treatment strategies for diabetic patients is the early identification of CVD risk factors, followed by the implementation of effective preventive measures.

Obesity, particularly central obesity, is a significant global public health concern [4]. Long-term research has demonstrated a strong correlation between obesity and an increased risk of CVD incidence and mortality [5,6]. Identifying individuals at high risk of obesity is crucial for preventing the development of CVD. While body mass index (BMI) is the most commonly used tool for diagnosing obesity [7], its limitations in accurately assessing visceral and subcutaneous fat distribution have become increasingly evident as research has evolved [8–10]. Recent studies have suggested that WWI is a superior predictor of obesity, particularly central obesity [11]. Research has shown that an elevated WWI is closely associated with various disorders, including diabetes [12], abdominal aortic calcification [13], hyperuricemia [14], cardiovascular disease [15], and stroke [16]. The WWI has a pronounced prognostic impact on the prevalence and mortality rates of CVD [17]. Therefore, the WWI is regarded as a refined, prospective, and more precise indicator for assessing obesity and related health risks.

Sedentary behavior, defined as activities that require minimal energy expenditure while awake, has become a common lifestyle in modern society [18,19]. Growing research demonstrates a significant association between sedentary behavior and a variety of chronic diseases [20], such as diabetes [21,22], cardiovascular disease [23], obesity [24], sarcopenia [25], and depression [26,27]. Given that sedentary behavior is a modifiable risk factor, addressing it is essential, as it has the potential to influence disease incidence and mortality significantly.

Therefore, we hypothesized that the association between WWI and CVD in diabetic patients may be mediated by sedentary behavior. The objectives of our investigation were to identify the relationship between WWI and CVD in diabetic patients and to evaluate the potential mediating role of sedentary behavior in this association. These findings may contribute to the formulation of targeted public health guidelines aimed at reducing the incidence of adverse cardiovascular outcomes in patients with diabetes through modifications in sedentary behavior.

## Materials and methods

### Data source

This study utilized data from the National Health and Nutrition Examination Survey(NHANES), a nationally representative cross-sectional survey conducted by the National Center for Health Statistics (NCHS) in the United States [28]. Since 1999, NHANES has been conducted in a 2-year cycle. For each cycle, new participants are randomly selected from the U.S. civilian noninstitutionalized population using a complex, multistage probability sampling design to ensure national representativeness. The survey was designed to gather information on the nutritional and health status of the American population. The participants' demographic, socioeconomic, dietary, and health-related data were collected through questionnaires, physical examinations, and laboratory tests. The study protocol was evaluated and approved by the NCHS Research Ethics Review Committee, and written informed consent was obtained from each participant [29]. The NHANES database is publicly accessible, which waived the need for ethical review for this study. Detailed statistical data are available at https://wwwn.cdc.gov/nchs/nhanes/default.aspx

### Selection criteria for study population

This study integrated data from seven NHANES cycles, which were conducted between 2007 and 2020. The criteria for the diagnosis of diabetes were as follows [30,31]: 1) self-reported physician-diagnosed DM; 2) current use of diabetes medication or insulin therapy to lower blood glucose level; 3) a glycated hemoglobin (HbA1c) level of ≥6.5%; 4) a fasting plasma glucose level of ≥7.0 mmol/L; or 5) a two-hour glucose (OGTT) mmol/L level of ≥11.1 mmol/L. Participants meeting any of these criteria were included in the analysis. The exclusion criteria were as follows: 1) age < 20 years; 2) pregnancy; 3) missing data on body weight, waist circumference, or sedentary time; and 4) invalid questionnaires regarding CVD history.

### Exposure variable: WWI and sedentary behavior

Sedentary behavior was the mediating variable in the current study. Sedentary behavior was assessed through an interview via the Global Physical Activity Questionnaire (GPAQ) [32,33]. This assessment was conducted by trained interviewers at a mobile examination center via a computer-assisted personal interview system. The participants were asked, "The following question concerns time spent sitting at school, at home, while traveling, or during social activities, including desk work, car or bus travel, reading, playing cards, watching TV, or using a computer. Please exclude sleep time. On a typical day, how much time do you usually spend sitting?" The responses were recorded in minutes. Individuals were classified as having sedentary behavior if their daily sedentary time was 360 minutes or more [34].

In this study, the WWI as the exposure variable. The formula for WWI is as follows: $WWI = \frac{Waist(cm)}{\sqrt{Weight(kg)}}$ [35]. The physical data, including body weight and waist circumference, were collected by trained technicians at the Mobile Examination Center (MEC). The complete methods, which include protocols, equipment, and quality control, are available at the following website:https://wwwn.cdc.gov/nchs/nhanes/continuousnhanes/manual.aspx.

### Outcomes variable: CVD

This study's outcome variable was CVD. According to prior research, CVD was defined as the presence of any of five self-reported cardiovascular outcomes: angina pectoris, coronary heart disease (CHD), congestive heart failure (CHF),

heart attack, and stroke [36,37]. The following questions were posed to all participants: "Has a doctor or other health professional ever told you that you have congestive heart failure/coronary heart disease/angina/heart attack/stroke?" An answer of "yes" to these questions indicated CVD.

## Covariates definition

During the household interviews, standardized questionnaires were used to collect data on age, sex, race, education level, marital status, poverty-income-ratio, smoking status, disease conditions, and medication use. The questionnaires were administered by professionally trained medical personnel. Race was categorized as Mexican American, non-Hispanic White, non-Hispanic Black, or other. There were three education levels: less than high school, high school or equivalent, and college or above. Marital status was categorized as either partnered or unpartnered. Smoking status was divided into two categories: never smoked and smoker. The poverty-income-ratio was categorized into three categories: 0--1.0, 1.0--3.0, and >3.0. The participants were classified into 4 alcohol consumption groups on the basis of the self-report survey results for these questionnaires, as described in previous studies [38]. These are classified as follows: 1) never drinker (had < 12 drinks in a lifetime); 2) mild alcohol user (≤ 1 drink per day for females, ≤ 2 drinks per day for males on average over the past year, or binge drinking (≥ 4 drinks/occasion for females, ≥ 5 drinks/occasion for males on 1 day per month); 3) moderate alcohol user (≤ 2 drinks per day for females, ≤ 3 drinks per day for males, or binge drinking on 2~5 days per month); and 4) heavy alcohol user (≥ 3 drinks per day for females, ≥ 4 drinks per day for males, or binge drinking ≥ 5 days per month). The Mobile Examination Center conducted medical examinations and subsequent laboratory assessments to obtain anthropometric measurements and biochemical parameters. The body mass index(BMI) was determined by dividing the weight in kilograms by the height in meters squared. The results were categorized as ≤ 25.0 kg/m², 25.0–30.0 kg/m², and >30.0 kg/m². Furthermore, fasting blood samples were examined at baseline for higher high-density lipoprotein levels, triglycerides, blood urea nitrogen, serum creatinine, serum albumin, serum uric acid, alanine aminotransferase, white blood cell count, and glycated hemoglobin.

## Missing value management

Variables with more than 15% missing values, including those related to systolic blood pressure, diastolic blood pressure, LDL, and insulin levels, were initially excluded by the researchers. For the study's outcome, mediation variable, and exposure variables, data with missing values were excluded. Multiple imputation methods were employed to impute other missing values. Using the "mice" package in R software, the random forest algorithm (trained on other non-missing variables) was implemented to perform multiple imputations.

## Statistical analysis

Means ± standard deviations were used for continuous variables that conformed to a normal distribution, while medians (25th percentile, 75th percentile) were used for continuous variables that did not conform to a normal distribution. The baseline differences in continuous variables were assessed via analysis of variance (ANOVA). One-way ANOVA was conducted for normally distributed continuous variables, whereas the Kruskal-Wallis test was employed for non-normally distributed continuous variables. Logistic regression analysis was used to assess the associations among WWI, sedentary behavior, and CVD in diabetic patients. To account for potential confounding, the logistic regression models were adjusted for age, sex, race, poverty-income-ratio, insulin use, marital status, hypertension, total cholesterol, blood urea nitrogen, serum albumin, and smoking status. Restricted cubic splines (RCS) were used to analyze the relationships among WWI, sedentary behavior, and CVD in diabetic patients. The findings were further stratified to consider the effects of sex, race, marital status, poverty income ratio, smoking status, glycosylated hemoglobin, and alcohol consumption. In addition, likelihood ratio tests were used to assess the interaction between WWI and subgroups. To test the robustness of our findings,

we performed a sensitivity analysis. First, multiple imputation was applied to variables with missing values, and the association between WWI and CVD risk in patients with diabetes was validated across 10 imputed complete datasets. Second, we deleted people older than 70 years to verify the association between WWI and CVD risk in patients with diabetes. Third, we additionally adjusted for dietary inflammatory index (DII), total physical activity (MET-minutes/week), medication adherence, education level, and poverty-income-ratio. Finally, we validated the association by stratifying WWI into tertiles. Mediation analyses were performed using the "Mediation" package in R software. The mediating effect of sedentary behavior on the association between WWI and diabetes-related CVD was assessed after adjusting for age, sex, race, poverty-income-ratio, insulin use, marital status, hypertension, total cholesterol, plasma urea nitrogen, serum albumin, and smoking status. The presence of a mediating effect was defined as meeting all of the following conditions: significant indirect effect, significant total effect, and positively proportional mediating effect. The data were analyzed via R software (version 4.3.2). P values <0.05 were considered statistically significant (two-tailed).

## Results

### Baseline characteristics and results analysis of the study participants

Between 2007 and 2020, the NHANES cycles identified a total of 8,703 non-pregnant diabetic patients aged 20 years or older. Of these, 3,047 participants were excluded due to missing sedentary time data, 110 due to missing cardiovascular outcome data, and 607 due to missing waist circumference or weight data. The final analytical sample consisted of 4,937 participants. A detailed screening process is illustrated in Fig 1 and S1 Data provides comprehensive information on the enrolled population.

Table 1 presents the differences in demographic and clinical characteristics between diabetic patients with and without CVD. Supplementary Table 1 in S1 File presents baseline characteristics of patients with T2DM by WWI quartiles. The

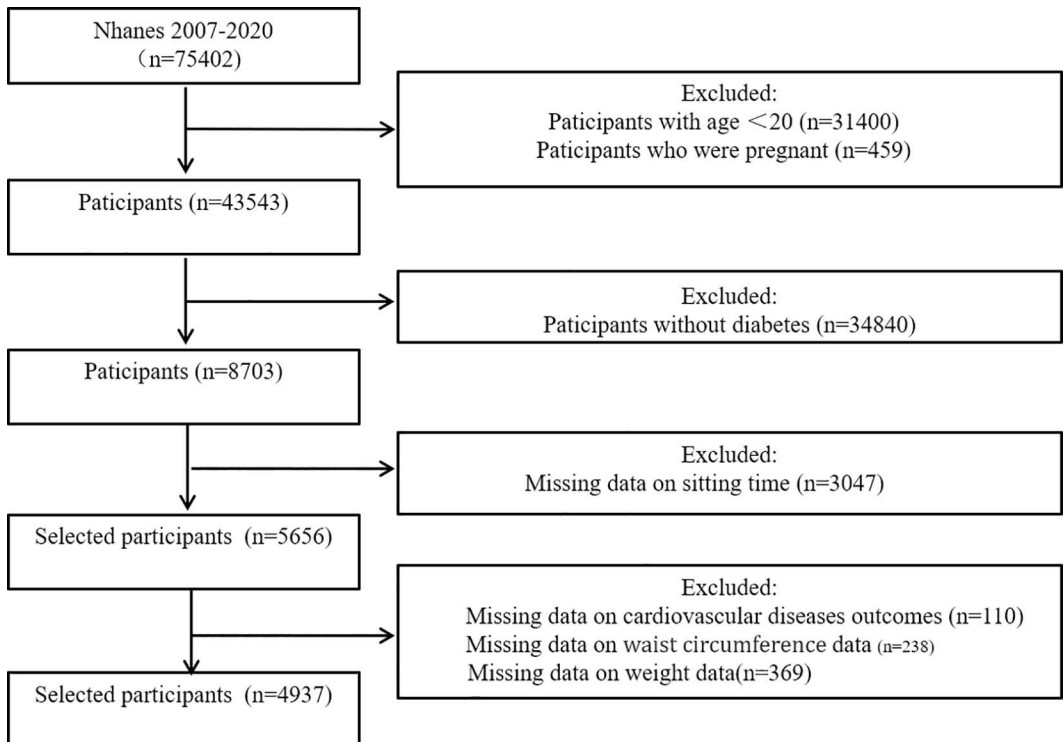

**Fig 1. Flow chart of the sample selection process.**

**Table 1. Baseline characteristics of the participants.**

| Characteristics | Overall(4937) | Non-CVD (3854) | CVD (1083) | p.overall |
|---|---|---|---|---|
| Age (years) | 62.0 (51.0,71.0) | 60.0 (49.0,68.0) | 68.0 (61.0,76.0) | <0.001 |
| Gender | | | | <0.001 |
| Female n (%) | 2401 (48.6%) | 1952 (50.6%) | 449 (41.5%) | |
| Male n (%) | 2536 (51.4%) | 1902 (49.4%) | 634 (58.5%) | |
| Race(%) | | | | <0.001 |
| Mexican American | 851 (17.2%) | 726 (18.8%) | 125 (11.5%) | |
| Other Hispanic | 571 (11.6%) | 468 (12.1%) | 103 (9.51%) | |
| Non-Hispanic white | 1662 (33.7%) | 1159 (30.1%) | 503 (46.4%) | |
| Non-Hispanic black | 1173 (23.8%) | 922 (23.9%) | 251 (23.2%) | |
| Other race | 680 (13.8%) | 579 (15.0%) | 101 (9.33%) | |
| Education(%) | | | | <0.001 |
| Less than high school | 786 (15.9%) | 612 (15.9%) | 174 (16.1%) | |
| High school diploma | 1860 (37.7%) | 1399 (36.3%) | 461 (42.6%) | |
| More than high school | 2291 (46.4%) | 1843 (47.8%) | 448 (41.4%) | |
| Marital status (%) | | | | <0.001 |
| Having a partner | 2997 (60.7%) | 2395 (62.1%) | 602 (55.6%) | |
| No partner | 1940 (39.3%) | 1459 (37.9%) | 481 (44.4%) | |
| PIR(%) | | | | 0.008 |
| < 1.0 | 1160 (23.5%) | 883 (22.9%) | 277 (25.6%) | |
| 1.0-3.0 | 2204 (44.6%) | 1702 (44.2%) | 502 (46.4%) | |
| ≥ 3.0 | 1573 (31.9%) | 1269 (32.9%) | 304 (28.1%) | |
| Smoking status(%) | | | | <0.001 |
| No | 2555 (51.8%) | 2106 (54.6%) | 449 (41.5%) | |
| Yes | 2382 (48.2%) | 1748 (45.4%) | 634 (58.5%) | |
| Alcohol intake status(%) | | | | <0.001 |
| Heavy | 618 (12.5%) | 503 (13.1%) | 115 (10.6%) | |
| Mild | 2513 (50.9%) | 1904 (49.4%) | 609 (56.2%) | |
| Moderate | 419 (8.49%) | 351 (9.11%) | 68 (6.28%) | |
| Nerver | 1387 (28.1%) | 1096 (28.4%) | 291 (26.9%) | |
| Insulin ues(%) | | | | <0.001 |
| No | 4026 (81.5%) | 3263 (84.7%) | 763 (70.5%) | |
| Yes | 911 (18.5%) | 591 (15.3%) | 320 (29.5%) | |
| Hypertension,(%) | | | | <0.001 |
| No | 1806 (36.6%) | 1596 (41.4%) | 210 (19.4%) | |
| Yes | 3131 (63.4%) | 2258 (58.6%) | 873 (80.6%) | |
| BMI(%) | | | | 0.052 |
| < 25 | 711 (14.4%) | 572 (14.8%) | 139 (12.8%) | |
| 25-30 | 1472 (29.8%) | 1166 (30.3%) | 306 (28.3%) | |
| ≥ 30 | 2754 (55.8%) | 2116 (54.9%) | 638 (58.9%) | |
| HDL(mmol/L) | 1.19(1.01,1.45) | 1.19 (1.01,1.47) | 1.14 (0.96,1.40) | <0.001 |
| BUN(mmol/L) | 5.00 (3.93,6.78) | 5.00 (3.93,6.43) | 6.07 (4.64,7.85) | <0.001 |
| Serum creatinine (mmol/L) | 76.9 (63.6,95.5) | 74.3 (61.2,90.2) | 88.4 (72.9,111) | <0.001 |
| Serum uric acid (mmol/L) | 333 (280,398) | 327 (274,393) | 357 (292,422) | <0.001 |
| Albumin(mg/L) | 41.0 (39.0,44.0) | 42.0 (39.0,44.0) | 41.0 (39.0,43.0) | <0.001 |
| ALT(U/L) | 27.1±27.1 | 27.6±19.9 | 25.3±43.9 | 0.102 |
| WBC(1000 cells/uL) | 7.62±2.41 | 7.61±2.42 | 7.68±2.38 | 0.39 |

*(Continued)*

**Table 1.** (Continued)

| Characteristics | Overall(4937) | Non-CVD (3854) | CVD (1083) | p.overall |
|---|---|---|---|---|
| TG (mmol/L) | 4.81±1.21 | 4.92±1.22 | 4.43±1.11 | <0.001 |
| HBA1C(%) | 7.18±1.76 | 7.19±1.80 | 7.16±1.60 | 0.593 |
| Sedentary time(hour) | 6.09±3.43 | 5.91±3.38 | 6.73±3.50 | <0.001 |
| WWI(cm/√kg) | 11.6±0.73 | 11.6±0.73 | 11.8±0.69 | <0.001 |
| Sedentary behavior(%) | | | | <0.001 |
| No | 2374 (48.1%) | 1953 (50.7%) | 421 (38.9%) | |
| Yes | 2563 (51.9%) | 1901 (49.3%) | 662 (61.1%) | |

PIR: Poverty-Income-Ratio; BMI: Body Mass Index; BUN: Blood Urea Nitrogen; ALB: Serum Albumin; HbA1c: Glycosylated Hemoglobin; TG: Total Cholesterol; WBC: White Blood Cell Count; HDL: Higher High-Density Lipoprotein; ALT: Alanine Aminotransferase; WWI: Weight-Adjusted Waist Index.

study included 4,937 patients with diabetes, of whom 1,083 had CVD. Among all participants, 2,536 (44.9%) were male, and the mean age was 62 years. The mean WWI was significantly higher in diabetic patients with CVD ($11.80\pm0.69 \frac{cm}{\sqrt{kg}}$) compared to those without CVD ($11.60\pm0.73 \frac{cm}{\sqrt{kg}}$). Diabetic patients with CVD were significantly older, more likely to be male, and had lower educational attainment and a lower poverty-income-ratio. They also exhibited higher levels of blood urea nitrogen, serum creatinine, and serum uric acid. Additionally, CVD patients had higher rates of hypertension, greater insulin use, longer sedentary time, and higher smoking prevalence. Conversely, they had higher high-density lipoprotein (HDL) levels and lower serum albumin levels. Notably, CVD patients also had higher rates of overweight and sedentary behavior.

## Association between WWI and CVD in diabetes

The association between WWI and CVD in patients with diabetes was analyzed using three logistic regression models, with progressive adjustments for confounding variables (Table 2). WWI was considered both a continuous and categorical

**Table 2.** Logistic regression analysis of the associations among WWI, sedentary time and CVD in patients with diabetes mellitus.

| Variable | Model 1 | | | Model 2 | | | Model 3 | | |
|---|---|---|---|---|---|---|---|---|---|
| | OR | OR 95% CI | P-value | OR | OR 95% CI | P-value | OR | OR 95% CI | P-value |
| WWI | 1.69 | 1.47-1.94 | <0.001 | 1.54 | `1.30-1.84 | <0.001 | 1.27 | 1.06-1.53 | <0.01 |
| WWI | | | | | | | | | |
| Q1 | Reference | | | Reference | | | Reference | | |
| Q2 | 1.74 | 1.35-2.25 | <0.001 | 1.49 | 1.13-1.95 | <0.01 | 1.6 | 1.23-2.07 | <0.001 |
| Q3 | 1.99 | 1.49-2.67 | <0.001 | 1.51 | 1.09-2.10 | <0.05 | 1.51 | 1.13-2.02 | <0.01 |
| Q4 | 2.94 | 2.22-3.90 | <0.001 | 2.41 | 1.74-3.34 | <0.001 | 2.24 | 1.63-3.09 | <0.001 |
| p for trend | | | <0.001 | | | <0.001 | | | <0.01 |
| Sedentary time | 1.07 | 1.04-1.09 | <0.001 | 1.06 | `1.03-1.09 | <0.001 | 1.04 | 1.02-1.07 | <0.01 |
| Sedentary time | | | | | | | | | |
| <6hour | Reference | | | Reference | | | Reference | | |
| ≥6hour | 1.53 | 1.30-1.81 | <0.001 | 1.37 | 1.13-1.65 | <0.001 | 1.28 | 1.06-1.55 | <0.01 |

Model 1: unadjusted.

Model 2: adjusted for age, gender, race.

Model 3: adjusted for multivariate variables: age, gender, race, insulin use, hypertension, total cholesterol, blood urea nitrogen, serum albumin, smoking status.

variable. When treated as a continuous variable, WWI was significantly associated with an increased risk of CVD in patients with diabetes. In Model 1 (unadjusted), WWI showed a significant positive association with CVD (OR = 1.69, 95% CI = 1.47–1.94, P < 0.001). This association persisted after adjusting for age, gender, and race in Model 2 (OR = 1.54, 95% CI = 1.30–1.84, P < 0.001) and after multivariable adjustment in Model 3 (OR = 1.27, 95% CI = 1.06–1.53, P < 0.01). Similarly, a dose-response relationship was observed across WWI quartiles (Q1-Q4), with the highest quartile (Q4) showing the strongest association in all models (P for trend <0.001 in all models).

## Association between sedentary behavior and CVD in diabetes

Sedentary time exhibited a significant association with CVD in patients with diabetes, regardless of whether it was analyzed as a continuous variable or categorized (Table 2). When analyzed as a continuous variable, increased sedentary time was significantly associated with increased CVD risk in all models (P < 0.001 in all models). In the fully adjusted categorical model, using the group with sedentary time <6 hours as the reference, individuals with sedentary time ≥6 hours had a significantly higher risk of CVD (OR = 1.28, 95% CI = 1.06–1.55, P < 0.01) in patients with diabetes. These findings highlight a robust positive association between prolonged sedentary behavior and CVD risk among patients with diabetes.

## Association between WWI and sedentary behavior in diabetes

The logistic regression analysis results for the association between WWI and sedentary behavior in patients with diabetes mellitus are presented in Table 3. The analysis was conducted using three models, with Model 1 being unadjusted, Model 2 adjusted for age, gender, and race, and Model 3 adjusted for multivariate variables including age, gender, race, insulin use, hypertension, total cholesterol, blood urea nitrogen, serum albumin, and smoking status. When WWI was analyzed as a continuous variable, we found that an increase in WWI was associated with an increase in sedentary behavior in diabetic patients (Model 1: OR = 1.28, 95% CI: 1.15–1.43; Model 2: OR = 1.37, 95% CI: 1.21–1.55; all P < 0.001). This association remained in the fully adjusted Model 3 (OR = 1.39, 95% CI: 1.23–1.58, P < 0.001). WWI as a categorical variable was significantly associated with increased sedentary behavior in the fourth quartile in the fully adjusted categorical model using the lowest quartile of WWI as the baseline, with an odds ratio (OR) of 1.89 (95% CI 1.41–2.54). This finding underscores the positive association between elevated WWI values and the likelihood of sedentary behavior in diabetic patients. (Table 3).

Table 3. Logistic regression analysis of the association between WWI and sedentary behavior in patients with diabetes mellitus.

| Variable | Model 1 | | | Model 2 | | | Model 3 | | |
|---|---|---|---|---|---|---|---|---|---|
| | OR | OR 95% CI | P-value | OR | OR 95% CI | P-value | OR | OR 95% CI | P-value |
| WWI | 1.28 | 1.15-1.43 | <0.001 | 1.37 | `1.21-1.55 | <0.001 | 1.39 | 1.23-1.58 | <0.01 |
| WWI | | | | | | | | | |
| Q1 | Reference | | | Reference | | | Reference | | |
| Q2 | 1.18 | 0.97-1.44 | 0.102 | 1.24 | 1.01-1.53 | <0.05 | 1.27 | 1.01-1.59 | <0.05 |
| Q3 | 1.38 | 1.11-1.71 | <0.01 | 1.45 | 1.16-1.81 | <0.01 | 1.47 | 1.16-1.86 | <0.01 |
| Q4 | 1.62 | 1.26-2.10 | <0.001 | 1.82 | 1.37-2.41 | <0.001 | 1.89 | 1.41-2.54 | <0.001 |
| p for trend | | | <0.001 | | | <0.001 | | | <0.01 |

Model 1: unadjusted.

Model 2: adjusted for age, gender, race.

Model 3: adjusted for multivariate variables: age, gender, race, insulin use, hypertension, total cholesterol, blood urea nitrogen, serum albumin, smoking status.

### RCS analysis

The RCS analysis further explored the associations between WWI, sedentary time, and CVD in patients with diabetes, as shown in Fig 2. In the fully adjusted model, RCS analysis revealed that both WWI and sedentary time exhibited approximately linear relationships with CVD risk in patients with diabetes (P for nonlinearity >0.05). Specifically, WWI and sedentary time were positively associated with CVD risk, as depicted in Fig 2A and 2B, respectively.

### Mediating effect of sedentary behavior on WWI and CVD in diabetes patients

Based on the mediation model adjusted for age, sex, race, poverty-income-ratio, insulin use, marital status, hypertension, total cholesterol, plasma urea nitrogen, serum albumin, and smoking status, the total effect of sedentary behavior at WWI on CVD in diabetes patients was significant (total effect, 0.0067, 95% CI, 0.003–0.008; P<0.001). Sedentary behavior partially mediated the association WWI and CVD in patients with diabetes (indirect effect, 0.0009; 95% CI, 0.0002–0.001; P<0.001). The proportion that was mediated was 13.43% (0.0009/0.0067). (Fig 3).

### Subgroup analysis analyses

We performed subgroup analyses stratified by sex, marital status, race, education level, poverty-income-ratio, smoking status, glycated hemoglobin, alcohol status, and BMI. The P-values for interaction were greater than 0.05 across all groups. These analyses demonstrated that the association between WWI and CVD remained stable across various subgroups (Fig 4).

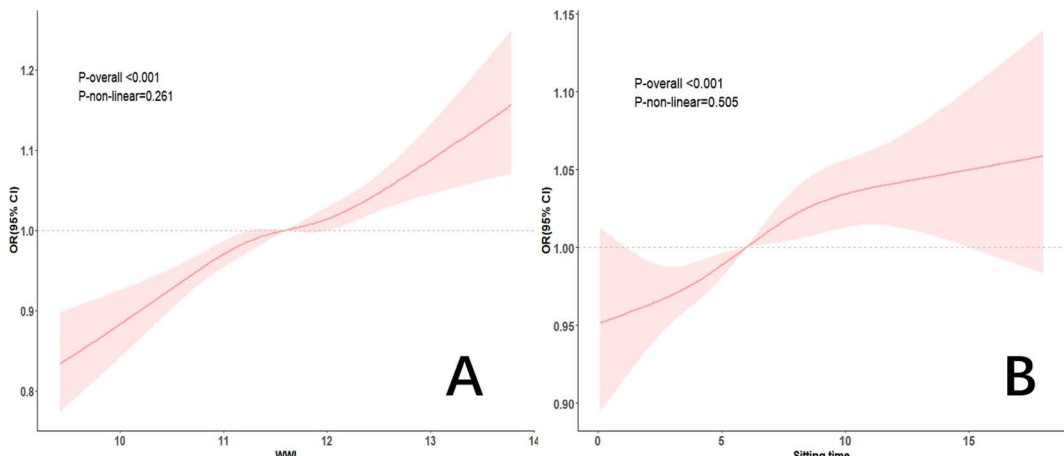

**Fig 2. RCS curves for WWI, sedentary behavior and CVD in patients with diabetes (A) Association between WWI and CVD in diabetes patients.** (B) Association between sedentary time and CVD in diabetes patients from NHANES 2007–2020.

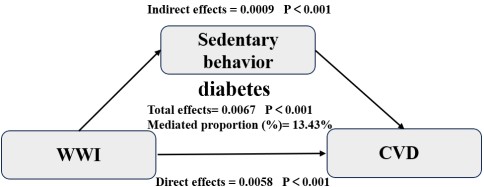

**Fig 3. Mediation models illustrating the relationship between WWI and CVD in patients with diabetes, with sedentary behavior as the mediator.**

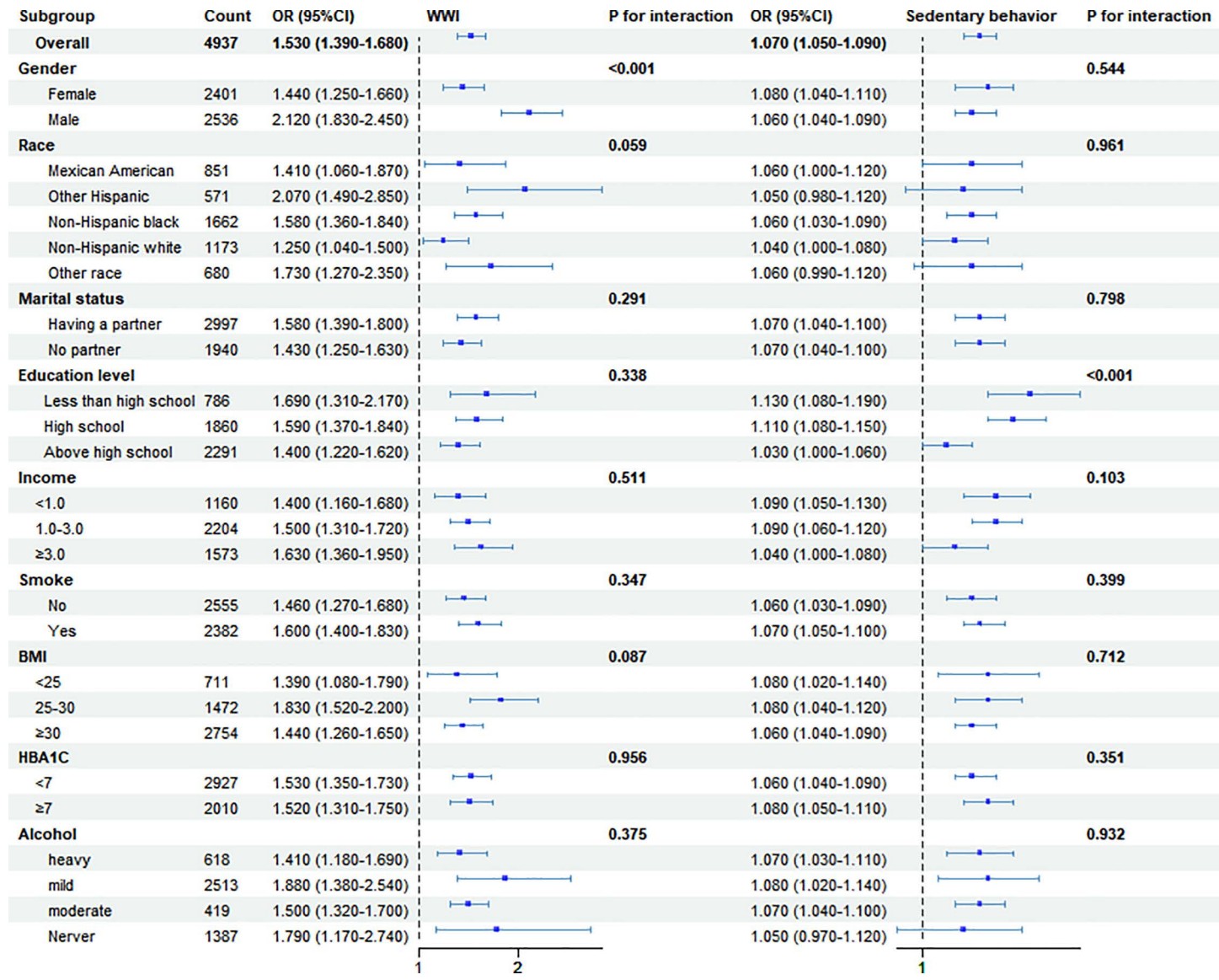

**Fig 4. Forest plot for the associations among WWI, sedentary behavior and CVD in patients with diabetes.**

## Sensitivity analyses

To assess the robustness of our findings, we conducted several sensitivity analyses. First, we evaluated the association between WWI and CVD in 10 imputed datasets to address potential bias from missing data. The results were consistent across all imputed datasets (Supplementary Figure 1 in S1 File), further supporting the robustness of our findings. Second, we repeated the analysis after excluding participants aged 70 years or older. Multivariable logistic regression confirmed that the association between WWI and CVD persisted, with results consistent with our initial findings (Supplementary Table 2 in S1 File). Third, additional adjustment for dietary inflammatory index (DII), physical activity (MET-minutes/week), medication adherence, education level, and poverty-income-ratio yielded essentially unchanged associations(Supplementary Table 3 in S1 File). Finally, participants in the highest WWI tertile exhibited a considerably elevated risk of CVD compared to those in the lowest tertile (Supplementary Table 4 in S1 File).

## Discussion

Our findings indicate that elevated WWI levels are positively associated with increased CVD risk in patients with diabetes, with similar trends observed for sedentary behavior. RCS analysis revealed a positive and approximately linear relationship between WWI and CVD risk in patients with diabetes. Mediation analysis further demonstrated that sedentary behavior accounted for 13.43% of the association between WWI and CVD, suggesting a partial mediating role. These results imply that strategies targeting sedentary behavior reduction may be effective in mitigating both central obesity and CVD risk in patients with diabetes.

Obesity, particularly central obesity, is associated with metabolic disorders such as insulin resistance and hyperinsulinemia [39], which are major contributors to diabetes [40] and recognized risk factors for CVD and mortality in adults [41]. In diabetic patients, cardiovascular complications are the leading causes of mortality and disability [42]. Thus, identifying simple and effective methods to assess obesity and determine CVD risk in diabetic individuals is a critical issue in clinical practice. The "obesity paradox" has been the subject of an expanding number of studies in recent years [43–46]. This phenomenon is characterized by individuals with overweight or obesity, as measured by BMI, having a lower all-cause mortality rate than those with normal weight [47,48]. This paradox has been observed in both general and high-risk populations. Similar outcomes have been reported when obesity is measured by WC [49]. The consensus is that this phenomenon arises because BMI and WC fail to distinguish between muscle mass and fat mass. To address this limitation, Park Y et al. introduced a novel obesity indicator termed the "WWI" [11]. In addition to diminishing the relationship between waist circumference and BMI, the WWI also incorporates the advantages of waist circumference, better reflects body fat distribution and muscle mass, and addresses the issue of central obesity, which is unrelated to weight [50,51]. In a study of 602 elderly individuals, Ansan et al. demonstrated that WWI correlates with adipose and muscle mass as measured by dual-energy X-ray absorptiometry, bioelectrical impedance analysis, and abdominal computed tomography [17]. These findings suggest that WWI may be a more accurate indicator for predicting abdominal adiposity in adults. WWI has been associated with various conditions, including fatty liver [52], cognitive function [53], hyperuricemia [14], diabetes [12], kidney stones [54,55], and urinary incontinence [56]. Additionally, studies have shown that WWI is positively associated with CVD [15], cardiovascular mortality, and all-cause mortality [50,57]. In this study, we included 4,937 adults with diabetes and found that WWI was positively associated with CVD risk, regardless of whether WWI was treated as a continuous or categorical variable.

Sedentary behavior, defined as any waking behavior with an energy expenditure of ≤1.5 metabolic equivalents while in a sitting, reclining, or lying position, is becoming increasingly prevalent in the population and is associated with various adverse health outcomes [58]. Extensive epidemiological studies have shown correlations between sedentary behavior and elevated risks of cardiovascular disease [59], diabetes [60,61], cancer [62], and overall mortality [63,64]. Recent evidence consistently indicates a strong association between sedentary behavior and the risk of cardiovascular diseases [65]. Our study extends these findings to diabetic populations, demonstrating a positive correlation between sedentary time and cardiovascular disease risk in this population. These results underscore sedentary behavior as a significant modifiable risk factor for cardiovascular disease in diabetes patients. Notably, our mediation analysis revealed that sedentary behavior mediated 13.43% of the association between WWI and cardiovascular disease (CVD), suggesting its partial intermediary role in this pathway. From a mechanistic perspective, this mediation may occur through several distinct pathways: First, sedentary behavior is strongly associated with metabolic disorders, including insulin resistance, dyslipidemia, and chronic inflammation, which are key contributors to CVD risk in diabetic patients [66]. Second, sedentary behavior may impair vascular endothelial function and induce hemodynamic changes that promote atherosclerosis [67]. Additionally, increased sympathetic nerve activity and blood pressure during prolonged sedentary periods may further exacerbate CVD risk [68]. These findings carry important clinical implications. Behavior interventions targeting sedentary time reduction could serve as an adjunctive strategy to weight management in obese diabetic patients. Particularly for patients struggling with weight loss, focusing initially on reducing and interrupting sedentary periods may provide cardiovascular protection while working toward long-term weight management goals.

A significant strength of our study is the use of a large and representative NHANES database, which enhances the generalizability of our findings. Our results provide robust evidence supporting the positive association between WWI and CVD risk in patients with diabetes. Furthermore, our study highlights the potential of WWI as an easily obtainable anthropometric index for identifying individuals at risk of CVD among diabetic patients. Given its straightforward calculation and low economic cost, WWI is an ideal option for healthcare institutions, particularly those with limited medical resources. Additionally, sedentary behavior, a modifiable risk factor, offers a pathway to mitigate CVD risk in diabetic patients with elevated WWI levels through early assessment and timely intervention.

However, there are several limitations in this study. First, as a cross-sectional study, our research can only demonstrate associations among WWI, sedentary behavior, and CVD development in people with diabetes but cannot establish causality. Future prospective studies and randomized controlled trials (RCTs) are needed to determine whether reducing sedentary behavior directly mitigates CVD risk or modifies WWI in this population. Second, the use of self-reported questionnaires to assess CVD and sedentary behavior has inherent limitations, including potential recall bias and inaccuracies in self-diagnosis. Therefore, further prospective randomized controlled trials are needed to verify the results of this study. While self-report is a practical method for large epidemiological studies, future research would benefit from incorporating objective measures such as medical record verification for CVD diagnoses and accelerometer data for sedentary behavior assessment. Third, while we comprehensively adjusted for potential confounders, residual confounding from unmeasured variables remains possible. Fourth, the study population was derived from U.S. databases, which may limit the generalizability of our findings to other ethnic groups or populations. Additionally, our research focused solely on diabetic patients; therefore, caution should be exercised when extrapolating these results to non-diabetic populations. Future multicenter studies involving diverse populations are warranted to validate our findings.

## Conclusion

The findings of this study demonstrate that an elevated WWI is associated with an increased risk of CVD in patients with diabetes. Furthermore, the results suggest that sedentary behavior may serve as a mediator in the positive association between WWI and CVD incidence. Thus, early clinical surveillance to identify individuals with a high WWI and the implementation of active weight management strategies may assist in mitigating CVD risk in individuals with diabetes.

## Supporting information

**S1 File. Supplementary file.**
(DOCX)

**S1 Data. Data used for analysis in the study.**
(XLSX)

## Author contributions

**Conceptualization:** Wei Xie, Guo Shao.

**Data curation:** Min Wu.

**Formal analysis:** Hongwei Zhu, Peng Wang, Yunqi Hua.

**Funding acquisition:** Hongwei Zhu, Guo Shao.

**Investigation:** Hongwei Zhu, Wei Xie, Guo Shao.

**Methodology:** Hongwei Zhu, Peng Wang, Guo Shao, Zhihui Li, Min Wu.

**Resources:** Guo Shao.

**Software:** Peng Wang, Zhihui Li.

**Supervision:** Hongwei Zhu.

**Validation:** Peng Wang.

**Writing – original draft:** Hongwei Zhu, Wei Xie, Yunqi Hua, Zhihui Li, Min Wu.

**Writing – review & editing:** Wei Xie, Guo Shao.

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
