## [Decision Letter · Decision Letter 0]

28 Jan 2025

Dear Dr. Zhu,

Thank you for submitting your manuscript to PLOS ONE. After careful consideration, we feel that it has merit but does not fully meet PLOS ONE’s publication criteria as it currently stands. Therefore, we invite you to submit a revised version of the manuscript that addresses the points raised during the review process.

**ACADEMIC EDITOR:**
**- The gap between what's already known and what you need to prove must be more clear in the introduction section**
**- The practical implication of your finding must be more detailed.**

We look forward to receiving your revised manuscript.

Kind regards,

Omnia Samir El Seifi, M.D.

Academic Editor

PLOS ONE

Journal Requirements:

-Is systemic inflammation a missing link between cardiometabolic index

with mortality? Evidence from a large population-based study (https://doi.org/10.1186/s12933-024-02251-w)

-The association between the weight-adjusted-waist index and frailty in US older adults: a cross-sectional study of NHANES 2007–2018 (https://doi.org/10.3389/fendo.2024.1362194)

(among others)

In your revision ensure you cite all your sources (including your own works), and quote or rephrase any duplicated text outside the methods section. Further consideration is dependent on these concerns being addressed.

“This work was supported by the National Natural Science Fund (82060337), the Shenzhen LongGang District Medical and Health Science and Technology Plan Project (LGKCYLWS2021000033, LGKCYLWS2023025), the Shenzhen City Science and Technology Plan Project Basic Research Surface Project (JCYJ20220531092412028, JCYJ20230807121306012), and the Baotou City Health Science and Technology Project (wsjkkj2022057).”

Reviewers' comments:

Reviewer's Responses to Questions

**Comments to the Author**

1. Is the manuscript technically sound, and do the data support the conclusions?

Reviewer #1: Yes

Reviewer #2: Yes

2. Has the statistical analysis been performed appropriately and rigorously?

Reviewer #1: Yes

Reviewer #2: Yes

3. Have the authors made all data underlying the findings in their manuscript fully available?

Reviewer #1: Yes

Reviewer #2: Yes

4. Is the manuscript presented in an intelligible fashion and written in standard English?

Reviewer #1: Yes

Reviewer #2: Yes

Reviewer #1: REVIEWER COMMENT AND SUGGETIONS DIABETIC

Generally congratulation to authors on writing this manuscript its interesting paper however there are several issues need improving to make it clear.

•Adhere to journal guideline on organizing the work

•Work extensively to be clear grammar and typographical errors throughout the document

ABSTRACT

On part of abstract the authors should sit and re write again because are not smart on part of background authors starting with the purpose of study (what new of your study or problem) also method important information are missing then on result the way authors presented mixing on line no 28 clear but line no 30 writing half also line no 25, 26 What is the meaning of the portion of conclusion that needs some improvement? Authors should conclude based on the results found. Some readers are interested in reading only the abstract before coming in, therefore. I suggest improving to make them more clear and understandable.

Method

•The authors should give more information of eligibility criteria and the sources and methods of selection of participants. Also describe methods of follow-up.

•Also the authors should define all outcomes

•And also should explain any effort address the potential source of bias

•Explain how quantitative approach was handled in the analysis

•On part of statistical methods the authors should give method used to control for confounding.

Result

•The authors should give the reasons for non-participation at each stage

•Consider use of a flow diagram

•I noted line no 168-174 it is ectopic revise and improve

•Also I noted line no 175-179 are not clear understanding what you presented the authors should be revised

•The table also need some of improvement are very feinted

•Line no 182 where it is.

•Line 183 the authors need to improve are not well smart

•The way the result of association present are not clear understood this part are crucial the authors need to improving and make it clear for the readers

Discussions

•The authors should revise this part and Summarise key results with reference to study objectives

Limitation

•I noted the authors should use scientific language of research on writing(our study is not without limitations) improve

Conclusions

•The authors should conclude according to the result.

Reference

•Several references do not fit the requirements of Vancouver style. Revise and improve them.

Reviewer #2: Good paper with valuable information. However there are some issues which needed to modify. First, I think that we already know that there is a relationship between body mass index and cardiovascular disease in patients with diabetes and sedentary life style. Could you explain more about novelty. Second, is the asking patients if they have cardiovascular disease the only assessment tool for diagnosing of cardiovascular disease? Is it enough?

**Do you want your identity to be public for this peer review?** For information about this choice, including consent withdrawal, please see our Privacy Policy

Reviewer #1: **Yes: ** rehema abdallah

Reviewer #2: **Yes: ** Laleh Abadi marand

---

## [Author Response · Author response to Decision Letter 1]

24 Apr 2025

Editor

Comments 1, “Please ensure that your manuscript meets PLOS ONE's style requirements, including those for file naming. The PLOS ONE style templates can be found at https://journals.plos.org/plosone/s/file?id=wjVg/PLOSOne_formatting_sample_main_body.pdf and

https://journals.plos.org/plosone/s/file?id=ba62/PLOSOne_formatting_sample_title_authors_affiliations.pdf.”

Response: According to the editor’s advice, we have revised the format of our manuscript to ensure compliance with PLOS ONE's Submission Guidelines.

Comments 2, “We noticed you have some minor occurrence of overlapping text with the following previous publication(s), which needs to be addressed:

-Is systemic inflammation a missing link between cardiometabolic index

with mortality? Evidence from a large population-based study (https://doi.org/10.1186/s12933-024-02251-w)

-The association between the weight-adjusted-waist index and frailty in US older adults: a cross-sectional study of NHANES 2007–2018 (https://doi.org/10.3389/fendo.2024.1362194)

(among others)

In your revision ensure you cite all your sources (including your own works), and quote or rephrase any duplicated text outside the methods section. Further consideration is dependent on these concerns being addressed.”

Response: Thanks for your comments. We have carefully reviewed the concerns regarding potential text overlap with previously published works and have thoroughly addressed these concerns in the revised manuscript. We have rephrased the content to ensure originality while preserving scientific accuracy and meaning. We believe these revisions have significantly improved the manuscript and resolved the issues related to text overlap.

Comments 3, “Thank you for stating the following financial disclosure:

“This work was supported by the National Natural Science Fund (82060337), the Shenzhen LongGang District Medical and Health Science and Technology Plan Project (LGKCYLWS2021000033, LGKCYLWS2023025), the Shenzhen City Science and Technology Plan Project Basic Research Surface Project (JCYJ20220531092412028, JCYJ20230807121306012), and the Baotou City Health Science and Technology Project (wsjkkj2022057).”

Please include this amended Role of Funder statement in your cover letter; we will change the online submission form on your behalf.”

Response: Thank you for your feedback. The funding mentioned above was provided by Professors Guo Shao and Hongwei Zhu, who were involved in the study design and manuscript revision. We have included this information in manuscript and their roles have been acknowledged in the revised manuscript.

Comments 4, “We note that your Data Availability Statement is currently as follows: All relevant data are within the manuscript and its Supporting Information files.

Please confirm at this time whether or not your submission contains all raw data required to replicate the results of your study. Authors must share the “minimal data set” for their submission. PLOS defines the minimal data set to consist of the data required to replicate all study findings reported in the article, as well as related metadata and methods (https://journals.plos.org/plosone/s/data-availability#loc-minimal-data-set-definition).�https://journals.plos.org/plosone/s/data-availability#loc-minimal-data-set-definition�.

https://journals.plos.org/plosone/s/recommended-repositories。

If there are ethical or legal restrictions on sharing a de-identified data set, please explain them in detail (e.g., data contain potentially sensitive information, data are owned by a third-party organization, etc.) and who has imposed them (e.g., an ethics committee). Please also provide contact information for a data access committee, ethics committee, or other institutional body to which data requests may be sent. If data are owned by a third party, please indicate how others may request data access.”

Response: Thank you for your feedback. In accordance with the editor’s advice, we have submitted all the original data required for replicating the research results in the supplementary materials. The data are detailed in lines 177-179 on page 9 and S1 Data.

Comments 5, “When completing the data availability statement of the submission form, you indicated that you will make your data available on acceptance. We strongly recommend all authors decide on a data sharing plan before acceptance, as the process can be lengthy and hold up publication timelines. Please note that, though access restrictions are acceptable now, your entire data will need to be made freely accessible if your manuscript is accepted for publication. This policy applies to all data except where public deposition would breach compliance with the protocol approved by your research ethics board. If you are unable to adhere to our open data policy, please kindly revise your statement to explain your reasoning and we will seek the editor's input on an exemption. Please be assured that, once you have provided your new statement, the assessment of your exemption will not hold up the peer review process.”

Response: Thank you for your comments. We have now provided all the raw data necessary for replicating the results of our study in S1 Data. This table contains the complete dataset utilized in our analysis, ensuring full transparency and reproducibility of our findings. Please refer to page 18 lines 349 and for details.

Comments 6, “Please review your reference list to ensure that it is complete and correct. If you have cited papers that have been retracted, please include the rationale for doing so in the manuscript text, or remove these references and replace them with relevant current references. Any changes to the reference list should be mentioned in the rebuttal letter that accompanies your revised manuscript. If you need to cite a retracted article, indicate the article’s retracted status in the References list and also include a citation and full reference for the retraction notice.”

Response: Thank you for your feedback. We have carefully reviewed our reference list to ensure its completeness and accuracy. We confirm that none of the references cited in our manuscript have been retracted.

Reviewer #1

Generally congratulation to authors on writing this manuscript its interesting paper however there are several issues need improving to make it clear.

Comments 1, “Adhere to journal guideline on organizing the work.”

Response: Thank you for your feedback. We have meticulously revised the manuscript to ensure full compliance with the guidelines of PLOS ONE."

Comments 2, “Work extensively to be clear grammar and typographical errors throughout the document.”

Response: Thank you for pointing this out. We have meticulously reviewed the entire manuscript, including all tables, figures, and captions, to ensure they are error-free and properly formatted. We are confident that these revisions have significantly enhanced the clarity and readability of the manuscript.

Comments 3, “ABSTRACT On part of abstract the authors should sit and rewrite again because are not smart on part of background authors starting with the purpose of study (what new of your study or problem) also method important information are missing then on result the way authors presented mixing on line no 28 clear but line no 30 writing half also line no 25, 26 What is the meaning of the portion of conclusion that needs some improvement? Authors should conclude based on the results found. Some readers are interested in reading only the abstract before coming in, therefore. I suggest improving to make them more clear and understandable.”

Response: Thank you for pointing this out. We have carefully considered the reviewer’s advice and meticulously revised the abstract to better align with the recommendations provided. Details of the revisions can be found on page 2, lines 17-38, highlighted in yellow in the revised manuscript.

Comments 4, “The authors should give more information of eligibility�criteria and the sources and methods of selection of participants. Also describe methods of follow-up.”

Response: Thank you for your feedback. In response, we have thoroughly revised the manuscript to provide more detailed information about the eligibility criteria, participant selection methods, and follow-up procedures. The eligibility criteria are detailed on page 5, lines 86-89; the participant selection methods are outlined on page 5, lines 84-92; and the methods of follow-up are described on page 4, lines 74-77. These sections have been meticulously reviewed and enhanced to ensure clarity and completeness.

Comments 5, “Also the authors should define all outcomes.”

Response: Thank you for your suggestion. We have meticulously revised the manuscript to provide clear definitions for outcome measures used in our study, based on established guidelines and definitions from the National Health and Nutrition Examination Survey (NHANES). These definitions can be found on page 6, lines 108-114, and are highlighted in the revised manuscript for easy reference.

Comments 6, “And also should explain any effort address the potential source of bias.”

Response: Thank you for pointing this out. In our study, we employed the NHANES survey, which utilizes a multistage probability sampling design to ensure representativeness and reduce selection bias. We adjusted for a comprehensive set of confounding variables in our analyses and conducted sensitivity analyses, such as excluding participants over the age of 70 and validating results across subgroups, to ensure the robustness and consistency of our findings. These measures were implemented to minimize potential biases and enhance the validity of our results. Details of these measures can be found on page 8, lines 150-163, and are highlighted in the revised manuscript for easy reference.

Comments 7, “Explain how quantitative approach was handled in the analysis.”

Response: Thank you for pointing this out. Descriptive statistics were summarized using means and standard deviations for normally distributed continuous variables, and medians with interquartile ranges for non-normally distributed continuous variables. Categorical variables were presented as percentages. Analysis of variance (ANOVA) and Kruskal-Wallis tests were employed to compare group differences. Multivariable logistic regression models were used to assess the associations between WWI, sedentary behavior, and cardiovascular disease (CVD) risk. Restricted cubic splines were utilized to explore potential nonlinear relationships and mediation analysis was conducted to evaluate the role of sedentary behavior in the WWI-CVD association. Sensitivity analyses, including subgroup validations and exclusion of participants over the age of 70, were performed to ensure the robustness of the findings. This systematic approach ensured a thorough and reliable analysis of the data. Details can be found on pages 7-8, lines 145-163, and are highlighted in the revised manuscript for easy reference.

Comments 8, “On part of statistical methods the authors should give method used to control for confounding.”

Response: Thank you for your comments. We used multivariable logistic regression models to assess the associations between WWI, sedentary behavior, and CVD risk. Based on the reviewer's feedback, we have conducted a sensitivity analysis to further validate the robustness of our findings. The specific results can be found in the S1 File. Stratified analyses were performed to assess the stability of the associations across different subgroups. These methods are consistent with established guidelines for controlling confounding in NHANES analyses and ensure that our results are robust and reliable. We have incorporated these details into the Methods section of our manuscript. Please refer to page 8 lines 150-163 and for details.

Comments 9, “Result The authors should give the reasons for non-participation at each stage.”

Response: Thank you for your suggestions. We have meticulously addressed this concern by revising the results section to provide detailed reasons for non-participation at each stage of the study. Details can be found on page 9, lines 174-179, and are highlighted in the revised manuscript for easy reference.

Comments 10, “Result: Consider use of a flow diagram.”

Response: Thank you for your suggestions. We have included a detailed flow diagram (Figure 1) that visually represents the participant selection process. Please see the Figure 1 on page 11.

Comments 11, “ I noted line no 168-174 it is ectopic revise and improve.”

Response: Thank you for pointing this out. Under the reviewer’s advice, we have thoroughly revised the relevant sentences. Details of the revisions can be found on page 9, lines 180-185, and are highlighted in the revised manuscript for easy reference.

Comments 12, “ Also I noted line no 175-179 are not clear understanding what you presented the authors should be revised.”

Response: Thank you for pointing this out. According to the reviewer’s advice, we have rewritten these sentences. Please refer to page 9 lines 185-191 and for details.

Comments 13, “Also The table also need some of improvement are very feinted.”

Response: Thank you for pointing this out. According to the reviewer’s advice, We have revised the table formatting to ensure consistency and readability. Please refer to Table1 on pages 9-10.

Comments 14, “ Line no 182 where it is.”

Response: Thank you for pointing this out. Thanks for your detailed review, I have corrected the errors and conducted a detailed review of the entire manuscript to improve readability.

Comments 15, “ Line 183 the authors need to improve are not well smart.”

Response: Thank you for pointing this out. According to the reviewer’s advice, we have made the necessary improvements. Please refer to page 11 lines 194 and for details.

Comments 16, “The way the result of association present are not clear understood this part are crucial the authors need to improving and make it clear for the readers.”

Response: Thank you for pointing this out. Following the reviewer’s advice, we have made the necessary improvements to enhance the clarity and accuracy of the manuscript. Details of these improvements can be found on pages 11-12, lines 194-213, and are highlighted in the revised manuscript for easy reference.

Comments 17, “Discussions: The authors should revise this part and Summarise key results with reference to study objectives.”

Response: Thank you for pointing this out. According to the reviewer’s advice, have carefully reviewed and revised this section to ensure that it clearly aligns with our study objectives and highlights the significance of our findings. Please see the discussion section on page 15.

Comments 18, “Limitation: I noted the authors should use scientific language of research on writing(our study is not without limitations) improve.”

Response: Thank you for pointing this out. According to the reviewer’s advice, we have rewritten these sentences. Please refer to page 17 lines 329 and for details.

Comments 19, “Conclusions The authors should conclude according to the result.”

Response: Thank you for your suggestion. According to the reviewer’s advice, we have rewritten these sentences. Please refer to page 18 lines342-346 and for details.

Comments 20, “Reference Several references do no

---

## [Decision Letter · Decision Letter 1]

14 May 2025

Dear Dr. Zhu,

Thank you for submitting your manuscript to PLOS ONE. After careful consideration, we feel that it has merit but does not fully meet PLOS ONE’s publication criteria as it currently stands. Therefore, we invite you to submit a revised version of the manuscript that addresses the points raised during the review process.

We look forward to receiving your revised manuscript.

Kind regards,

Omnia Samir El Seifi, M.D., Ph.D.

Academic Editor

PLOS ONE

Journal Requirements:

Reviewers' comments:

Reviewer's Responses to Questions

**Comments to the Author**

Reviewer #1: All comments have been addressed

Reviewer #2: All comments have been addressed

2. Is the manuscript technically sound, and do the data support the conclusions?

Reviewer #1: Yes

Reviewer #2: Yes

3. Has the statistical analysis been performed appropriately and rigorously?

Reviewer #1: Yes

Reviewer #2: Yes

4. Have the authors made all data underlying the findings in their manuscript fully available?

Reviewer #1: Yes

Reviewer #2: Yes

5. Is the manuscript presented in an intelligible fashion and written in standard English?

Reviewer #1: Yes

Reviewer #2: Yes

Reviewer #1: Reviewer comment and suggestions

Generally this study presents a valuable exploration of the relationship between the weight-adjusted waist indexes (WWI), sedentary behavior, and cardiovascular disease (CVD) risk among patients with diabetes. Here are some points to be improved regarding the methodology, results, and conclusions:

Strengths:

Sample Size: The study utilizes data from a substantial sample (4,937 participants) across multiple NHANES cycles, which enhances the generalizability of the findings.

Analytical Approach: The use of logistic regression models and restricted cubic spline analyses is appropriate for assessing the associations between WWI, sedentary behavior, and CVD risk. Additionally, subgroup and sensitivity analyses strengthen the robustness of the results.

Significant Findings: The study successfully identifies a positive association between WWI and CVD risk, as well as the role of sedentary behavior, contributing meaningful insights into how obesity and lifestyle factors interplay in diabetic patients.

Weaknesses:

Causality: While the study identifies associations, it does not establish causality. Longitudinal studies or randomized control trials may be needed to confirm whether reducing sedentary behavior directly decreases CVD risk or WWI in diabetic patients.

Sedentary Behavior Measurement: The study may benefit from clarifying how sedentary behavior is measured (e.g., self-reported vs. objective measures) and whether the measurement tools have been validated. This aspect is crucial as misreporting can lead to bias.

Potential Confounding Variables: Although the study accounts for confounders, it remains unclear if all relevant factors (e.g., diet, physical activity levels, medication adherence, and socioeconomic status) were adequately controlled. Further elaboration on this is necessary.

Generalizability: The focus on diabetic patients limits the generalizability of the findings to the broader population, including people without diabetes. Furthermore, the study’s findings should be interpreted cautiously across different ethnic and age groups.

Interpretation of Mediation Analysis: The conclusion regarding sedentary behavior mediating 13.43% of the association between WWI and CVD should be elaborated. The significance and clinical implications of such a percentage could be discussed further, especially regarding how it translates to actual practice or lifestyle changes.

Overall, this study adds important knowledge about the interrelationships between obesity, sedentary behavior, and cardiovascular health in diabetic patients. However, further longitudinal research and clearer methodologies regarding measurement and confounding factors are necessary to strengthen the conclusions and implications for clinical practice. Future work could focus on intervention studies that aim to reduce sedentary behavior as a means of lowering CVD risk among diabetic populations.

Reviewer #2: Thank you, the paper is completed and All of the comments have been addressed. Details of limitations and the novelty have been explained.

**Do you want your identity to be public for this peer review?** For information about this choice, including consent withdrawal, please see our Privacy Policy

Reviewer #1: No

Reviewer #2: **Yes: ** Laleh Abadi marand

---

## [Author Response · Author response to Decision Letter 2]

24 Jul 2025

Editor

Comments, “Please review your reference list to ensure that it is complete and correct. If you have cited papers that have been retracted, please include the rationale for doing so in the manuscript text, or remove these references and replace them with relevant current references. Any changes to the reference list should be mentioned in the rebuttal letter that accompanies your revised manuscript. If you need to cite a retracted article, indicate the article’s retracted status in the References list and also include a citation and full reference for the retraction notice.”

Response: Thank you for your feedback. We have carefully reviewed our reference list to ensure its completeness and accuracy. We confirm that none of the references cited in our manuscript have been retracted.

Reviewer #1

Reviewer comment and suggestions

Generally this study presents a valuable exploration of the relationship between the weight-adjusted waist indexes (WWI), sedentary behavior, and cardiovascular disease (CVD) risk among patients with diabetes. Here are some points to be improved regarding the methodology, results, and conclusions:

Strengths:

Sample Size: The study utilizes data from a substantial sample (4,937 participants) across multiple NHANES cycles, which enhances the generalizability of the findings.

Analytical Approach: The use of logistic regression models and restricted cubic spline analyses is appropriate for assessing the associations between WWI, sedentary behavior, and CVD risk. Additionally, subgroup and sensitivity analyses strengthen the robustness of the results.

Significant Findings: The study successfully identifies a positive association between WWI and CVD risk, as well as the role of sedentary behavior, contributing meaningful insights into how obesity and lifestyle factors interplay in diabetic patients.

Comments 1, “While the study identifies associations, it does not establish causality. Longitudinal studies or randomized control trials may be needed to confirm whether reducing sedentary behavior directly decreases CVD risk or WWI in diabetic patients.”

Response: Thank you for your comments. We acknowledge the inherent limitations of our cross-sectional study design in establishing causal relationships. To address this point, we have added the following statement to the Limitations section: “As a cross-sectional study, our research can only demonstrate associations among WWI, sedentary behavior, and CVD development in people with diabetes, rather than establish causality. Future prospective studies and randomized controlled trials (RCTs) are needed to determine whether reducing sedentary behavior directly mitigates CVD risk or modifies WWI in this population. ” Details can be found on page 18, lines 335-339, and are highlighted in the revised manuscript for easy reference.

Comments 2, “Sedentary Behavior Measurement: The study may benefit from clarifying how sedentary behavior is measured (e.g., self-reported vs. objective measures) and whether the measurement tools have been validated. This aspect is crucial as misreporting can lead to bias.”

Response: Thank you for your comments. We recognize that using self-reported questionnaires to assess sedentary behavior has inherent limitations, including potential recall bias and inaccuracies in self-reported. Although this method is not as precise as objective measures, it is widely employed in large-scale population studies like NHANES due to its practicality and cost-effectiveness (Association between daily sitting time and kidney stones based on the National Health and Nutrition Examination Survey (NHANES) 2007-2016: a cross-sectional study. Int J Surg 110, 4624-4632, 2024; The association of sedentary behaviour and physical activity with periodontal disease in NHANES 2011-2012. J Clin Periodontol 49, 758-767,2022). However, we acknowledge that this approach may introduce limitations, and we have incorporated these into the limitations section of our manuscript. Details can be found on page 18, lines 339-345, and are highlighted in the revised manuscript for easy reference.

Comments 3, “Potential Confounding Variables: Although the study accounts for confounders, it remains unclear if all relevant factors (e.g., diet, physical activity levels, medication adherence, and socioeconomic status) were adequately controlled. Further elaboration on this is necessary.”

Response: Thank you for your comments. In the revised manuscript, we have added a sensitivity analysis that further adjusts for: dietary inflammatory index, total MET-minutes per week, prescription medication adherence, and Socio-economic status (education level and the NHANES family poverty-to-income ratio). After adjusting for these covariates, the associations remained unchanged, indicating that the result was robust (Supplementary Table 4). The corresponding methods and results have been updated; details can be found on page 8, lines 163–165 and pages 14-15, lines 265–267 highlighted in yellow for easy reference.

Comments 4, “Generalizability: The focus on diabetic patients limits the generalizability of the findings to the broader population, including people without diabetes. Furthermore, the study’s findings should be interpreted cautiously across different ethnic and age groups.”

Response: Thank you for your comments. We have added the following statement to the Limitations section:“ the study population was derived from U.S. databases, which may limit the generalizability of our findings to other ethnic groups or populations. Additionally, our research focused solely on diabetic patients; therefore, caution should be exercised when extrapolating these results to non-diabetic populations. Future multicenter studies involving diverse populations are warranted to validate our findings.” Details can be found on page 18, lines 347-350, and are highlighted in the revised manuscript for easy reference.

Comments 5, “Interpretation of Mediation Analysis: The conclusion regarding sedentary behavior mediating 13.43% of the association between WWI and CVD should be elaborated. The significance and clinical implications of such a percentage could be discussed.”

Response: Thank you for your comments. We thank the reviewer for this valuable suggestion. In response, we have expanded our discussion of the mediation effect to better highlight its clinical significance. While this represents partial mediation, it suggests sedentary behavior contributes meaningfully to the association between WWI and CVD in diabetes. We have added a discussion of three key mechanistic pathways (metabolic dysregulation, vascular dysfunction, and sympathetic activation) and emphasized that targeting sedentary behavior could serve as a practical adjunct to weight management, particularly for patients struggling with weight loss. These additions appear in the Discussion section and underscore how even modest reductions in sedentary time may offer cardiovascular benefits in this high-risk population. Details can be found on pages 16-17, lines 308-325, and are highlighted in the revised manuscript for easy reference.

Comments 6, “Overall, this study adds important knowledge about the interrelationships between obesity, sedentary behavior, and cardiovascular health in diabetic patients. However, further longitudinal research and clearer methodologies regarding measurement and confounding factors are necessary to strengthen the conclusions and implications for clinical practice. Future work could focus on intervention studies that aim to reduce sedentary behavior as a means of lowering CVD risk among diabetic populations.”

Response: Thank you for your comments. We have: (1) explicitly addressed the need for longitudinal designs to establish causality in our Limitations section; (2) strengthened methodological rigor through comprehensive adjustment for key confounders (including DII, physical activity, medication adherence, and socioeconomic factors) with stable results in sensitivity analyses; and (3) expanded discussion of clinical implications to highlight sedentary behavior reduction as a practical intervention target for diabetic patients. While we agree that future intervention studies will be valuable to confirm these findings, our study provides important epidemiological evidence characterizing the interrelation ships between WWI, sedentary behavior, and CVD risk in diabetes, addressing a significant gap in current research. These revisions have enhanced the study's scientific rigor while maintaining its clinical relevance for diabetes management.

---

## [Decision Letter · Decision Letter 2]

13 Aug 2025

Sedentary behavior mediates the association between weight-adjusted waist index and cardiovascular disease in patients with diabetes

PONE-D-24-46590R2

Dear Dr. Zhu,

We’re pleased to inform you that your manuscript has been judged scientifically suitable for publication and will be formally accepted for publication once it meets all outstanding technical requirements.

Kind regards,

Omnia Samir El Seifi, M.D., Ph.D.

Academic Editor

PLOS ONE

Additional Editor Comments (optional):

Reviewers' comments:

Reviewer's Responses to Questions

**Comments to the Author**

Reviewer #1: All comments have been addressed

2. Is the manuscript technically sound, and do the data support the conclusions?

Reviewer #1: Partly

3. Has the statistical analysis been performed appropriately and rigorously?

Reviewer #1: Yes

4. Have the authors made all data underlying the findings in their manuscript fully available?

Reviewer #1: Yes

5. Is the manuscript presented in an intelligible fashion and written in standard English?

Reviewer #1: Yes

Reviewer #1: Reviewer comment and suggestions

This study investigates the relationship between the weight-adjusted waist index (WWI), sedentary behavior, and cardiovascular disease (CVD) in patients with diabetes, using data from 4,937 participants across NHANES cycles (2007–2020). The key findings are:

•Patients with diabetes and CVD had higher WWI levels compared to those without CVD.

•WWI was positively associated with CVD risk, even after adjusting for confounders (OR=1.27, 95% CI: 1.06–1.53).

•Sedentary behavior was independently associated with increased CVD risk (OR=1.04, 95% CI: 1.02–1.07).

•Both WWI and sedentary time showed linear relationships with CVD risk.

•Mediation analysis indicated that sedentary behavior mediates approximately 13.4% of the association between WWI and CVD.

Strengths of the study include:

•Large, nationally representative sample.

•Use of multiple analytical methods, including logistic regression, spline models, and mediation analysis.

•Consideration of confounding factors and subgroup analyses to verify robustness.

Limitations and considerations:

•Cross-sectional design limits causal inference; temporal relationships cannot be confirmed.

•Sedentary behavior and WWI are both modifiable factors, but intervention studies are needed to assess causal effects.

•The mediation percentage (13.4%) suggests other unmeasured pathways also contribute to the WWI-CVD relationship.

•The measurement of sedentary time may rely on self-report, which can introduce bias.

Implications:

•Highlighting WWI as a potential marker of CVD risk in diabetic patients.

•Emphasizing the importance of reducing sedentary behavior could help mitigate obesity-related CVD risk.

•Further longitudinal or interventional research is warranted to determine causality and effective strategies.

Overall, this study contributes valuable insights into the complex interplay between obesity indices, lifestyle behaviors, and cardiovascular health in diabetes, underlining the importance of managing sedentary habits alongside obesity reduction efforts.

**Do you want your identity to be public for this peer review?** For information about this choice, including consent withdrawal, please see our Privacy Policy

Reviewer #1: **Yes: ** rehema abdallah

---

## [Editor Report · Acceptance letter]

PONE-D-24-46590R2

PLOS ONE

Dear Dr. Zhu,

I'm pleased to inform you that your manuscript has been deemed suitable for publication in PLOS ONE. Congratulations! Your manuscript is now being handed over to our production team.

Kind regards,

on behalf of

Professor Omnia Samir El Seifi

Academic Editor

PLOS ONE